# COVID-19 Vaccine Uptake and Associated Factors in Sub-Saharan Africa: Evidence from a Community-Based Survey in Tanzania

**DOI:** 10.3390/vaccines11020465

**Published:** 2023-02-17

**Authors:** Sia E. Msuya, Rachel N. Manongi, Norman Jonas, Monica Mtei, Caroline Amour, Melina B. Mgongo, Julieth S. Bilakwate, Maryam Amour, Albino Kalolo, Ntuli Kapologwe, James Kengia, Florian Tinuga, Frida Ngalesoni, Abdalla H. Bakari, Fatimata B. Kirakoya, Awet Araya, Innocent B. Mboya

**Affiliations:** 1Institute of Public Health, Kilimanjaro Christian Medical University College, Moshi P.O. Box 2240, Tanzania; 2Community Health Department, KCMC Hospital, Moshi P.O. Box 3010, Tanzania; 3Internal Medicine Department, Faculty of Medicine, Kilimanjaro Christian Medical University College, Moshi P.O. Box 2240, Tanzania; 4London School of Hygiene and Tropical Medicine, Keppel Street, London WC1E 7HT, UK; 5Department of Community Health, Muhimbili University of Health and Allied Sciences, Dar es Salaam P.O. Box 65032, Tanzania; 6Department of Public Health, St. Francis University College of Health and Allied Sciences, Morogoro P.O. Box 175, Tanzania; 7President’s Office-Regional Administration and Local Government, Dodoma P.O Box 1923, Tanzania; 8Immunization and Vaccine Development Program, Ministry of Health, Dodoma P.O. Box 743, Tanzania; 9Amref Health Africa in Tanzania, Dar es Salaam P.O. Box 2773, Tanzania; 10School of Natural Science, The State University of Zanzibar, Tunguu P.O. Box 146, Tanzania; 11United Nations Children’s Fund (UNICEF), Dar es Salaam P.O. Box 4076, Tanzania; 12Department of Translational Medicine, Lund University, 214 28 Malmö, Sweden

**Keywords:** COVID-19 infection, vaccine hesitancy, vaccine uptake, general community, Tanzania

## Abstract

COVID-19 is a major public health threat associated with the increased global burden of infectious diseases, mortality, and enormous economic loss to countries and communities. Safe and efficacious COVID-19 vaccines are crucial in halting the pandemic. We assessed the COVID-19 vaccine uptake and associated factors among community members from eight regions in Tanzania. The interviewer-administered questionnaire collected data. Multiple logistic regression models determined the factors associated with vaccine uptake. The median age of 3470 respondents was 37 years (interquartile range of 29–50 years) and 66% of them were females. Only 18% of them had received the COVID-19 vaccine, ranging from 8% in Dar es Salaam to 37% in Simiyu regions. A third (34%) of those vaccinated people did not know which vaccine they were given. Significantly higher rates of COVID-19 vaccine uptake were among the respondents aged 30+ years, males, and with a history of COVID-19 infection. Unfavorable perceptions about vaccine safety and efficacy lowered the rates of vaccine uptake. Setting-specific interventions and innovations are critical to improving vaccine uptake, given the observed differences between regions. Efforts are needed to increase vaccine uptake among women and younger people aged less than 30 years. Knowledge-based interventions should enhance the understanding of the available vaccines, benefits, target groups, and availability.

## 1. Introduction

The COVID-19 pandemic is a public health problem globally [1]. Vaccination against COVID-19 is one of the key and lifesaving preventive measures against the COVID-19 pandemic [2,3,4,5]. Studies of COVID-19 vaccines have shown significant effects such as reducing severe infections, hospitalization, and deaths [5,6]. Vaccinating at least 70% or more of the population is needed to create herd immunity and reduce the severity of the disease and mortality, thus halting the epidemic [3,7,8]. While there were multiple vaccines in development in November 2021, WHO approved seven vaccines to be used including Pfizer–BioNTech, Oxford/AstraZeneca, Sinopharm, Moderna, Janssen, Sinovac-CoronaVac, and Bharat Biotech BBV152 Covaxin [4]. The development and delivery of safe and effective COVID-19 vaccines have yielded remarkable results [4]. As of 16 November 2022, over 12.9 million COVID-19 vaccine doses had been administered globally, with the African region having the lowest coverage rates, with only 24.1% of the population being fully vaccinated [8,9,10].

Tanzania, contrary to other East African countries, introduced the COVID-19 vaccines late [5,11,12]. Due to differing political responses, the introduction of COVID-19 vaccines in Tanzania was met with intense debates on different platforms (social media and other media channels) on its usefulness and safety [5,11,13]. Myths and misconceptions and fear of the negative effects were widely shared, fueled by the initial denial from some religious and political leaders, hence there was increased vaccine hesitancy [11,12,13,14]. In addition, vaccine uptake and hesitancy have also been associated with social-demographic characteristics such as age (low uptake in young people), sex (low uptake among females), perceived risk of infection, low education level, low socio-economic status, negative attitudes towards vaccines, among other factors [2,3,8,11,13,15,16]. The COVID-19 pandemic disrupted the delivery of routine healthcare services due to increased hospitalization, increased death rates, and mental health problems such as depression, anxiety, panic, and stress [4,17,18,19], in addition to economic and socio-cultural consequences such as the unprecedented disruptions of lifestyles, familial, social and cultural communications, and relations [18,20].

When the first batch of vaccines arrived on 24 July 2021 [21], the Tanzanian government prioritized healthcare workers, people with comorbidities, adults above 50 years of age, ports of entry workers, military and security forces, and school teachers [5]. Due to perceived high demand, in August 2021, the Tanzanian government changed the policy, and everyone aged 18 years and above could receive the COVID-19 vaccine voluntarily both in the mainland and in Zanzibar. According to the World Health Organisation, by 18 November 2022, the country had administered over 32.9 million doses of the vaccine, totaling nearly half of its population [9].

A recent review reported Tanzania to be among other countries with no data on COVID-19 vaccine acceptance rates [22]. Data on the levels and determinants (including facilitators and barriers) of the COVID-19 vaccine uptake among general community members in Tanzania are necessary to guide community mobilization campaigns by the government and other stakeholders. The current study sought to bridge this gap, in addition to providing relevant data to shape the design of tailored health education and promotion messages to increase vaccine uptake and coverage rates in the Tanzanian general community. The study assessed the awareness, knowledge, and risk perception of COVID-19 infection and vaccines, determined the prevalence of COVID-19 vaccine uptake and hesitancy, and investigated its associated factors among the general community members in Tanzania. 

## 2. Methods

### 2.1. Design, Setting, and Population

We carried out a community-based cross-sectional study in six (6) regions in the Tanzanian mainland and two (2) in Zanzibar between December 2021 and April 2022, when there was the highest vaccine hesitancy among general community members in Tanzania. The regions were selected based on the vaccine wastage rates as of 21 September 2021, whereby the regions with high vaccine wastage were the worst-performing ones, while those with low rates were better performing in terms of vaccine uptake. Two good, medium, and poor performing regions were selected considering zonal representation. The regions selected in the mainland were Dar es Salaam (eastern zone), Lindi (southern zone), Kilimanjaro (northern zone), Simiyu (lake zone), Tabora (western zone), and Mbeya (southern West Highlands zone). In Zanzibar, Pemba Kaskazini (In Pemba) and Mjini Magharibi (in Unguja) were the study sites. Two districts from each region were included considering rural–urban representativeness, hence there were sixteen districts in total. The study population was general community members across the eight selected regions. Specifically, the general community members included all adult men and women aged 18 years and above who were available at the household during data collection and provided informed consent.

### 2.2. Sample Size and Sampling

The sample size calculation for the quantitative survey was conducted using a single proportion formula given as [N = [(Z_α/2_)^2^ × p(1 − p)]/ε^2^, where N is the minimum required sample size, Z_α/2_ is the standard normal value corresponding to the 95% level of confidence, p is the hypothesized proportion of vaccine hesitancy in the general Tanzanian community, and ε is the margin of error. We used a standard normal value of 1.96 under the 95% confidence limit, assumed a 50% proportion of vaccine hesitancy among the general community members, and added a 20% non-response proportion and a 3% margin of error. The sample size was then multiplied by a design effect of two (2) to allow for a complex survey design, yielding 2600 participants. Furthermore, the data collection in the first three regions (Dar es Salaam, Mjini Magharibi, and Pemba Kaskazini) revealed lower uptake of the COVID-19 vaccine than the proportion used in the estimation of sample size. Therefore, the total sample size was increased by 35% in the remaining five regions, yielding a total of 3470 general community members. The multistage sampling technique selected participants in the quantitative survey. 

### 2.3. Data Collection

Face-to-face interviews were used to collect data from the community members using pre-tested questionnaires using the KoBoToolbox software. The questionnaire was developed based on previous studies to assess vaccine hesitancy and acceptability [13,14,23,24,25]. Trained research assistants (RAs) collected data in the local Swahili language. Data collection followed ethics approval and permission from the required bodies at the regional, district, and ward levels. On the day of data collection, the research teams met at the ward leader’s offices and were linked with the community health worker (CHW) of the respective ward. The CHWs introduced the researchers to the household members. At the village/street level, the first household was selected randomly (around the village/street office), and RAs continued with data collection in a circular motion to ensure representativeness. At the household level, the household head was invited to participate. If they were not around, another adult fulfilling the inclusion criteria was invited to participate. Where there were multiple residential households in one house, only two were included. Then, the RA introduced the study and its aims to the eligible participants and obtained verbal consent before the interviews. Interviews were conducted in the participants’ households (quiet corners, where quietness and privacy were sought) and lasted for 30–40 min. Daily debriefing identified and resolved issues encountered in the field. 

### 2.4. Data Analysis

The collected data were transferred from KoBoToolbox to an excel spreadsheet. Statistical Package for Social Sciences (SPSS) version 26 managed the data, and later transferred it to STATA version 15 for cleaning and analysis. The data analysis included descriptive and inferential statistics. The main outcome variable was the uptake of COVID-19 vaccines, which was measured by asking the participants; “Have you taken the COVID-19 vaccine?”, and the responses were “yes, will wait for some time, and will not take the vaccine at all”. Frequencies and proportions were used to summarize categorical variables and measures of central tendency and dispersion to summarize the numerical variables. The Chi-square test was used to compare COVID-19 vaccine uptake proportion by age, sex, region, education level, socio-economic status, perceived risk of COVID-19 infection, and history of COVID-19 infection. Statistically significant results were judged at a 5% level (*p* < 0.05). 

Household socio-economic status (SES) or wealth index/quintile was measured using questions assessing the ownership of household assets similar to those reported in the TDHS 2015/16 [26]. To generate SES, a latent variable was first predicted using a generalized structural equation model with the items about as predictors. Then, the latent variable was used to predict the wealth quintile or SES equally divided into five groups: lowest, second, middle, fourth, and highest wealth quintiles. The variable was summarized as frequencies and percentages. 

A composite measure of COVID-19 vaccines safety and efficacy perceptions were generated from 11 Likert scale items with responses ranging from “Strongly agree (1)” to “Strongly disagree (5)”, see Appendix A (Section V, question 38). Six items (items c, f, g, i, j, and k) were reverse-coded to ensure higher scores represented unfavorable perceptions. Cronbach’s alpha statistic assessed the reliability (internal consistency) of the 11 items, yielding a value of 76.1%. The sum of scores ranged from 11 to 55. Average scores (0.52) were obtained by dividing all of the scores by the maximum (55) and were used to categorize participants into two groups: from the minimum value to 0.52 reflected favorable perceptions/attitudes, and they were unfavorable, if otherwise. We also performed a sensitivity analysis on a continuous scale, i.e., perceptions of COVID-19 safety and efficacy score on the odds of vaccine uptake. The chi-squared test compared the proportions of COVID-19 vaccine uptake by perceptions of vaccine safety and efficacy (a binary variable). 

A multivariable logistic regression model determined the factors associated with vaccine uptake among the community members in this study. This model estimated the odds ratios (OR) and 95% confidence intervals (CIs). In the crude/unadjusted analysis, a logistic regression model was fit to each selected independent variable, i.e., age categories (years), education level, wealth quintile (SES), perceived risk of COVID-19 infection, history of COVID-19 infection, and perceptions towards vaccine safety and efficacy. Manual stepwise regression was used to select variables to retain in the multivariable (adjusted) analysis model that resulted in removing the education level. It is important to note here that education level was also used to compute the wealth quintile (SES). Stepwise regression (forward selection and backward elimination) procedures were also implemented to test the effect of self-reported history of co-morbidities on vaccine uptake, and we found no statistically significant effect. Additionally, for all of the regression analyses (except for the stepwise procedure that considered *p* < 0.1 for inclusion of variables in the adjusted analyses models), statistically significant results were judged at the 5% level (*p* < 0.05). 

## 3. Results

### 3.1. Participant Background Characteristics

The study included a total of 3503 general community members in the 16 districts, of whom 3470 (99.1%) consented to participate. The ages of 3470 participants ranged from 18–91 years, with a median age of 37 and an interquartile range of 29–50 years, and 66% were females. Of the 3470 participants, 71.2% were married/cohabiting, about 65% had primary/ no education, 27% were farmers, 32.7% conducted business activities, and 85.3% did not have health insurance (Table 1). 

### 3.2. Awareness, Knowledge, and Risk Perception of COVID-19 Infection and Vaccines

Nearly 23% of Tanzanians do not think COVID-19 exists. Only 26% of the 3470 participants perceived their risk for COVID-19 to be high, while the rest of them reported a very low or low risk (52%), and 22% did not know their risk. About 4% and 3% of the participants reported having a family member who was ever infected and died due to COVID-19 disease, respectively (Table 2).

Regarding COVID-19 vaccines, 94% of the 3470 participants heard about the vaccines, but the majority (75%) could not mention any vaccine by name. Upon probing, 28% could mention the Janssen vaccine. Thirty-six percent of all participants did not know where to obtain the COVID-19 vaccine. Nearly one in five participants (17%) reported that a person can become infected by having the COVID-19 vaccination, and about 40% did not know the key advantages of the COVID-19 vaccines, as only 58.8% mentioned it reducing the risk/chance of being infected with COVID-19, 60% mentioned it preventing severe infection, and 58.5% mentioned it reducing the number of deaths due to COVID-19 (Table 2).

### 3.3. Prevalence of COVID-19 Vaccine Uptake

Overall, 18% of the 3470 participants received the COVID-19 vaccine. The findings revealed significant differences in the prevalence of COVID-19 vaccine uptake by age, sex, region, education level, wealth quintile (SES), COVID-19 risk perception, and history of COVID-19 infection (Table 3).

The prevalence was highest (28.4% and 19.1%) among adults aged 50+ years and 40–49 years, respectively, and it was higher in males (22.2%) compared to that in females (16%), those in the Simiyu (37.3%) and Kilimanjaro (32%) regions, those with no education (21.7%), and those in the lowest wealth quintile (25.1%). The prevalence was also high among community members who perceived themselves as at high/very high risk of COVID-19 infection (23.7%), followed by those perceived to be at no or low risk of COVID-19 infection (19.1%). Nearly forty percent (38.8%) of those with a self-reported history of COVID-19 infection had been vaccinated. Those with a favorable perception of vaccine safety and efficacy had a higher proportion of vaccine uptake (28.7%) than those with unfavorable perceptions (9.5%) (Table 3).

### 3.4. Factors Associated with COVID-19 Vaccine Uptake

Adjusted for other factors, older participants (30+ years) were most likely to be vaccinated than their younger counterparts were. For every year of age increase, the odds of COVID-19 vaccine uptake increased by 3% (OR = 1.03, 95%CI 1.02, 1.04, *p* < 0.001) (results not in the table). Participants aged 50+ years had over three times higher odds of vaccine uptake than those aged <30 years (OR = 3.33, 95%CI 2.52, 4.40). Females were less likely to be vaccinated than males were (OR = 0.77, 95%CI 0.63, 0.94), even after adjusting for other factors. In addition, by adjusting for other factors, COVID-19 vaccine uptake differed significantly by region, whereby the participants from the Simiyu region had over eleven times (OR = 11.42, 95%CI 7.98, 16.35) higher odds than those in the Dar es Salaam region did. The odds of vaccine uptake were also high in Zanzibar, that is (OR = 1.75, 95% CI 1.20, 2.58) in Mjini Magharibi and (OR = 4.47, 95%CI 3.01, 6.65) in Pemba Kaskazini. Those at a low/very low perceived risk of COVID-19 infection had lower odds of vaccine uptake. In addition, having a self-reported history of COVID-19 infection increased the odds of vaccine uptake (OR = 2.45, 95%CI 1.37, 4.40). There were significantly lower odds of vaccine uptake among the community members with unfavorable perceptions about the COVID-19 vaccine safety and efficacy (OR = 0.22, 95%CI 0.17, 0.27) (Table 4). The results from an analysis of the perceptions of COVID-19 vaccine safety and efficacy score as a continuous variable agree with those for the binary variable reported in Table 4. By adjusting for other factors, the odds of vaccine uptake decreased by 14% (OR = 0.86, 95%CI 0.85, 0.88) for every increase in the perceptions of COVID-19 vaccine safety and efficacy score (results not in the table).

## 4. Discussion

We aimed to assess the COVID-19 vaccine uptake rate and its associated factors among the general community members in Tanzania. The study was conducted between December 2021 and April 2022, when there was the highest vaccine hesitancy among the general community members in Tanzania associated with issues around mistrust, safety, and a lack of reliable information about COVID-19 vaccines [2,11,16]. The study found that as of April 2022, only 18% of the general Tanzanian community had received the COVID-19 vaccine, ranging from 8% in Dar es Salaam to 37% in Simiyu regions. More than a third (34%) of those who were vaccinated did not know which vaccine they were given. The country has made remarkable progress to increase the vaccination coverage rates, vaccinating about half of its population [9,27]. Tanzania’s upward trends in COVID-19 vaccination coverage have been influenced by a sustained collaboration between the government and its partners [27].

In this study, significantly higher odds of COVID-19 vaccine uptake were among those aged 30+ years, males, and with a history of COVID-19 infection. Previous studies across Africa and low- and middle-income countries associated low vaccine uptake with younger age, females, and a low socio-economic status [15,16]. We found no statistically significant differences in vaccine uptake by education level and wealth quintile in this study. The findings emphasize that women need targeted messages because they have a lower uptake rate. Efforts are also needed to increase the vaccine uptake rate among younger people aged <30 years in Tanzania.

Evidence from two systematic reviews and meta-analyses demonstrated that good knowledge and positive attitudes about COVID-19 and vaccination have a significant impact on virus prevention and vaccine uptake across the globe [24,25]. Less than a quarter of participants in this study believed that COVID-19 did not exist in Tanzania, while just over half (51.6%) perceived themselves to have no or be at low risk of COVID-19 infection. In addition, although the majority of them (94%) had heard about COVID-19 vaccines, three-quarters of them could not mention any vaccine by name. As also reported elsewhere [13,16], unfavorable perceptions about vaccine safety and efficacy lowered the odds of uptake. These findings demonstrate sub-optimal community knowledge of COVID-19 and vaccines as a key preventive measure. Interventions to address COVID-19 vaccine acceptability and uptake in Tanzania should aim to improve the knowledge of the available vaccines, how they work, their advantages, who can be vaccinated, and where people can obtain the vaccines. Such interventions should go in hand with providing appropriate information to healthcare providers and community stakeholders to educate the public on the vaccines [5,25,28].

We also observed regional variations in COVID-19 vaccine uptake in the country. The findings suggest that identifying and resolving region-specific bottlenecks to vaccination coverage will increase uptake/coverage rates in rural and urban areas. Health promotion interventions should enhance the community’s knowledge of the benefits of COVID-19 vaccines, clearly communicating their short- and long-term side effects. The WHO in the African region reported the success of the strategy of assigning specific regions to technical and financial partners and has contributed to maintaining the upward trend in COVID-19 vaccination coverage [27]. Considering the socio-demographic, behavioral, economic, and geographical factors affecting uptake, COVID-19 vaccine hesitancy must therefore be addressed with context-specific interventions focused on the population groups with low uptake rates using different media platforms [3,5,8,11,13,25,28]. In addition, such interventions should go in hand with effective stakeholder engagement and health system strengthening [25,26].

This study has contributed to the literature on the current status of COVID-19 vaccine uptake/coverage in SSA, unlike previous publications that discuss intentions to uptake the vaccines if they become available [29]. Such data are essential to the governments and stakeholders such as the UN organizations, such as WHO, to see which areas they need to support in the country to improve the levels of COVID-19 vaccine uptake [23]. To the best of our knowledge, the study is one of its kind in Tanzania, as it assesses COVID-19 vaccine uptake and associated factors by taking a representative sample at the general community level using eight regions in the mainland and Zanzibar. Nevertheless, the study has several limitations. Firstly, since the study was conducted during a period of high vaccine hesitancy in the country (between December 2021 and April 2022), social desirability bias may have overestimated the vaccination uptake status, which was self-reported and was without validation from the participant’s vaccination certificate. Secondly, the proportion of vaccine uptake might have been overestimated because the participants might have reported having been vaccinated due to stigma and fear of disclosing their vaccination status. The study did not also ask for proof of vaccination. Thirdly, in the initial phase of the survey, the community members hesitated to participate in the survey in one ward especially because the link person was a street leader who was part of the ward COVID-19 response committee, thus they thought that the research team was conducting a mobilization campaign for vaccine uptake. Nevertheless, the community health workers were effective in community linkage because they are highly trusted by the local communities in Tanzania.

## 5. Conclusions

COVID-19 vaccine uptake is low in Tanzania. Setting-specific interventions and innovations are critical to improving vaccine uptake, given the observed differences between the regions. The campaign messages to increase vaccine uptake should be tailored to women and younger people aged less than 30 years. Knowledge-based interventions should enhance the understanding of the available vaccines and how they work, their advantages, who can be vaccinated, and where people can be vaccinated. Such interventions should address the reasons for hesitancy to promote vaccine uptake. Accurate and timely vaccine information coupled with a rapid response by the government and other stakeholders is essential in empowering community members to make informed choices about COVID-19 vaccine uptake. Qualitative studies should explore the barriers and facilitators of COVID-19 vaccination in order to inform community engagement and education programs.

## Figures and Tables

**Table 1 vaccines-11-00465-t001:** Participants’ background characteristics (N = 3470).

Variable	Frequency	Percentage
**Region**		
Dar es Salaam	1215	35.0
Kilimanjaro	275	7.9
Lindi	237	6.8
Mbeya	474	13.7
Simiyu	410	11.8
Tabora	313	9.0
Mjini Magharibi	327	9.4
Pemba Kaskazini	219	6.3
**Area of residence reported**		
Urban	2224	64.1
Rural	1246	35.9
**Age (years)**		
<30	966	27.8
30–39	929	26.8
40–49	669	19.3
50+	906	26.1
**Sex**		
Male	1167	33.6
Female	2303	66.4
**Education level**		
None	364	10.5
Primary	1877	54.1
Secondary	992	28.6
College/University	237	6.8
**Occupation**		
No job	222	6.4
Business	1136	32.7
Employed	1193	34.4
Farmer	919	26.5
**Marital status**		
Single	574	16.5
Married/Cohabiting	2470	71.2
Separated/Divorced	426	12.3
**Religion**		
None	52	1.5
Muslim	1790	51.6
Christian	1628	46.9
**Have health insurance**		
No	2959	85.3
Yes	511	14.7
**Wealth quintile**		
Lowest	704	20.3
Second	665	19.2
Middle	700	20.2
Fourth	702	20.2
Highest	699	20.1

**Table 2 vaccines-11-00465-t002:** Awareness, knowledge, and risk of COVID-19 infection among the general community members in Tanzania (N = 3470).

Variable	Frequency	Percentage
**Think that COVID-19 exists in Tanzania**		
No	287	8.3
Yes	2669	76.9
Do not know	514	14.8
**Perceived risk of COVID-19 infection**		
Not at all	740	21.3
Low/very low	1051	30.3
Medium-very high	911	26.3
Don’t know	768	22.1
**Ever been infected with COVID-19**		
No	3403	98.1
Yes	67	1.9
**Family member ever been infected with COVID-19**		
No	3319	95.6
Yes	151	4.4
**Family member ever died of COVID-19**		
No	3366	97.0
Yes	104	3.0
**Ever heard about COVID-19 vaccines**		
No	216	6.2
Yes	3254	93.8
**Vaccines heard of (n = 3254)**		
Don’t know the vaccines	2430	74.7
Janssen	894	27.5
Pfizer-BioNTech	137	4.2
Sinovac-CoronaVac	116	3.6
Sinopharm	104	3.2
Oxford/AstraZeneca	68	2.1
None	68	2.1
Moderna	54	1.7
Novavax	9	0.3
Sputnik V	8	0.2
Sputnik Light	5	0.2
**Know where to get COVID-19 vaccination**		
No	1251	36.1
Yes	2219	63.9
**Regarding COVID-19 vaccines ***		
A person can get infected by COVID-19 vaccination	586	16.9
A vaccinated person can get infected with COVID-19	1443	41.6
COVID-19 vaccine can be given to people who had been infected/ sick with COVID-19	1538	44.3
A person who has received the COVID-19 vaccine needs to continue following traditional preventive methods	2434	70.1
COVID-19 vaccines can reduce the risk/ chance of being infected with COVID-19	2039	58.8
COVID-19 vaccines prevent severe infection	2081	60.0
COVID-19 vaccines reduce deaths due to COVID-19	2030	58.5

* Frequencies and percentages among those who answered ‘Yes’.

**Table 3 vaccines-11-00465-t003:** Prevalence of vaccine uptake by selected characteristics among the general community members in Tanzania (N = 3470).

Variables	Total	Vaccinated (%)	*p*-Value
**Age (years)**			<0.001
<30	966	102 (10.6)	
30–39	929	140 (15.1)	
40–49	669	128 (19.1)	
50+	906	257 (28.4)	
**Sex**			<0.001
Male	1167	259 (22.2)	
Female	2303	368 (16.0)	
**Region**			<0.001
Dar es Salaam	1215	102 (8.4)	
Kilimanjaro	275	88 (32.0)	
Lindi	237	57 (24.1)	
Mbeya	474	69 (14.6)	
Simiyu	410	153 (37.3)	
Tabora	313	48 (15.3)	
Mjini Magharibi	327	54 (16.5)	
Pemba Kaskazini	219	56 (25.6)	
**Education level**			0.03
None	364	79 (21.7)	
Primary	1877	354 (18.9)	
Secondary	992	152 (15.3)	
College/University	237	42 (17.7)	
**Wealth quintile**			<0.001
Lowest	704	177 (25.1)	
Second	665	109 (16.4)	
Middle	700	105 (15.0)	
Fourth	702	106 (15.1)	
Highest	699	130 (18.6)	
**Perceived risk of COVID-19 infection**			<0.001
No/low risk	1791	342 (19.1)	
Medium risk	561	97 (17.3)	
High/very high risk	350	83 (23.7)	
Don’t know	768	105 (13.7)	
**Ever been infected with COVID-19**			<0.001
No	3286	591 (18.0)	
Yes	67	26 (38.8)	
Don’t know	117	10 (8.5)	
**Perceptions of COVID-19 vaccine safety and efficacy**			<0.001
Favorable	1104	445 (28.7)	
Unfavorable	1739	182 (9.5)	

**Table 4 vaccines-11-00465-t004:** Factors associated with COVID-19 vaccine uptake among the general community members in Tanzania (N = 3470).

Variables	COR (95%CI)	*p*-Value	AOR (95%CI)	*p*-Value
**Age (years)**				
<30	1.0		1.0	
30–39	1.50 (1.14, 1.97)	0.003	1.41 (1.05, 1.90)	0.022
40–49	2.00 (1.51, 2.66)	<0.001	1.75 (1.28, 2.37)	<0.001
50+	3.35 (2.61, 4.31)	<0.001	3.33 (2.52, 4.40)	<0.001
**Sex**				
Male	1.0		1.0	
Female	0.67 (0.56, 0.80)	<0.001	0.77 (0.63, 0.94)	0.01
**Region**				
Dar es Salaam	1.0		1.0	
Kilimanjaro	5.13 (3.71, 7.10)	<0.001	4.22 (2.95, 6.02)	<0.001
Lindi	3.46 (2.41, 4.95)	<0.001	3.00 (1.99, 4.53)	<0.001
Mbeya	1.86 (1.34, 2.58)	<0.001	2.43 (1.70, 3.47)	<0.001
Simiyu	6.50 (4.89, 8.64)	<0.001	11.42 (7.98, 16.35)	<0.001
Tabora	1.98 (1.37, 2.86)	<0.001	1.95 (1.32, 2.91)	0.001
Mjini Magharibi	2.16 (1.51, 3.08)	<0.001	1.75 (1.20, 2.58)	0.004
Pemba Kaskazini	3.75 (2.60, 5.40)	<0.001	4.47 (3.01, 6.65)	<0.001
**Education level**				
None	1.0			
Primary	0.84 (0.64, 1.10)	0.21	-	-
Secondary	0.65 (0.48, 0.88)	0.01	-	-
College/University	0.78 (0.51, 1.18)	0.24	-	-
**Wealth quintile**				
Lowest	1.0		1.0	
Second	0.58 (0.45, 0.76)	<0.001	0.97 (0.70, 1.33)	0.82
Middle	0.53 (0.40, 0.69)	<0.001	0.83 (0.60, 1.15)	0.26
Fourth	0.53 (0.41, 0.69)	<0.001	0.96 (0.69, 1.33)	0.79
Highest	0.68 (0.53, 0.88)	0.003	1.28 (0.91, 1.81)	0.16
**Perceived risk of COVID-19 infection**				
No/low risk	1.0		1.0	
Medium risk	0.89 (0.69, 1.14)	0.34	1.40 (1.06, 1.84)	0.02
High/very high risk	1.32 (1.00, 1.73)	0.05	1.14 (0.86, 1.53)	0.36
Don’t know	0.67 (0.53, 0.85)	0.001	0.84 (0.60, 0.16)	0.29
**Ever been infected with COVID-19**				
No	1.0		1.0	
Yes	2.89 (1.76, 4.76)	<0.001	2.45 (1.37, 4.40)	0.003
Don’t know	0.43 (0.22, 0.82)	0.01	0.71 (0.35, 1.45)	0.35
**Perceptions of COVID-19 vaccine safety and efficacy**				
Favorable	1.0		1.0	
Unfavorable	0.26 (0.22, 0.31)	<0.001	0.22 (0.17, 0.27)	<0.001

## Data Availability

The dataset that was used in this study is available upon request to the corresponding author.

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
