# Peer review of "COVID-19 Vaccine Uptake and Associated Factors in Sub-Saharan Africa: Evidence from a Community-Based Survey in Tanzania"

_vaccines, 2023, doi:10.3390/vaccines11020465_

Round 1

Reviewer 1 Report

Overall comments

COVID-19 is perhaps the most important global health challenge at the moment. Vaccines that are very safe and highly effect exist, but sadly vaccine uptake is lower than desired. This is true not only in the rich countries of the developed world, but also in low and middle income countries, where vaccination against COVID-19 may be even more important because of the relative unavailability of expensive and resource-intensive intensive care for the sickest patients. Understanding the epidemiology of vaccine hesitancy in many settings is the first step to enhancing vaccination rates. This manuscript describes the epidemiology of COVID-19 vaccine uptake in Tanzania, which has had a relatively problematic vaccine introduction due to various social and political controversies. The manuscript identifies relatively less vaccinated subpopulations, and pinpointing some demographic characteristics and concerns of people associated with vaccine hesitancy.

Minor concerns

In the Introduction the vaccines are listed my manufacturer, not by name, or both with the manufacturer in parentheses.

There are error messages generated at lines 172, 179,185, 221, .

“Jansen” as the relevant division of Johnson and Johnson is misspelled.

There is a spacing problem at line 194.

Introduction

The Introduction offers a good overview of the main concerns and motivations for the study, placing the study in the context of the larger problem of vaccine hesitancy and the COVID-19 pandemic.

Methods

The methods are well-described and reasonable for the goals of the study.

Is Swahili the only language used by the population studied? Is it universally understood? Did the investigators make an effort to determine whether the participants could understand the questions in the language used in the study?

Results

The manuscript should report the numbers for the wealth and education quintiles.

Conclusions/Discussion

In general, the conclusions are well-supported by the data.

Was messaging or outreach different in some way in the communities that had better vaccine coverage rates?

Figures and Tables

Table 1 needs to be clearer. The list of Frequency and Percentage are presumably the number of participants in each category and the percentage of the total number of participants in that category, but this should be clarified.

Tables 3 and 4. What is the reference group for the P-values.

Reviewer 2 Report

The topic is certainly worthy of attention. The focus was identified clearly, and the content was sufficiently organized and explained. The main strength of the study is that it covers a gap in knowledge in this field using a formally correct methodology, despite the limitations of a cross-sectional design and the use of self-reported data. However, I think it has some major and minor flaws. Furthermore, although the content was certainly understandable, the writing could be improved in some passages (abstract in particular).

The survey's primary problem is that it took a long time to conduct (about five months). It probably would have been preferable to attempt to complete the survey administration in a narrower window of time to examine a continuously changing and quickly evolving problem like hesitation against anti-COVID-19 vaccinations. If only to guarantee the data's internal consistency.

Another problem of greater significance is methodological. It concerns the categorization of participants into "favorable" and "unfavorable" to vaccination based on their responses to the 11 items on their COVID-19 vaccines safety and efficacy perceptions. Granted that this is a simplification, my main perplexity concerns the definition of the threshold score based on the mean score. If the test sample had an overall "favorable" opinion and the average score was very high, could we still have defined a subject who scored slightly below the average as "unfavorable" to vaccination?

In addition, although the results are significant and interesting, the discussion and conclusions contain considerations that are not directly supported by your findings or even by literature references. Especially those regarding concerns about the effects of vaccination on the product of conception and fertility.

Finally, the manuscript was very superficially prepared. Some bibliographic references are missing and there are several formatting errors.
